# Hydrothermal Synthesis of Layered Titanium Phosphate Ti$_2$O$_2$H(PO$_4$)[(NH$_4$)$_2$PO$_4$]$_2$ and Its Potential Application in Cosmetics

**Irina V. Kolesnik** [1,2,*], **Andrey N. Aslandukov** [1], **Anatoly S. Arkhipin** [1] and **Daniil A. Kozlov** [1,2]

1   Faculty of Materials Science, Lomonosov Moscow State University, Moscow 119991, Russia
2   Kurnakov Institute of General and Inorganic Chemistry of the Russian Academy of Sciences, Moscow 119991, Russia
*   Correspondence: kolesnik@inorg.chem.msu.ru

**Abstract:** Titanium phosphates were recently revealed as promising cosmetic pigments; however, their photocatalytic activity and sun protective factor (SPF) levels have not been investigated in detail. In this study, we used hydrothermal conditions to prepare nanocrystalline anatase, brookite, and layered titanium phosphate using the titanium lactate complex, NH$_4$H$_2$PO$_4$, and urea as precursors. The samples were characterized by powder X-ray diffraction (XRD) in addition to Raman spectroscopy, transmission and scanning electron microscopy (TEM, SEM), energy-dispersive X-ray spectroscopy (EDX), and UV-Vis spectroscopy. Furthermore, the photocatalytic activity, sun protective factor, and moisture retention ability were determined for the samples. Brookite exhibited the highest SPF value and anatase the lowest, while Ti$_2$O$_2$H(PO$_4$)[(NH$_4$)$_2$PO$_4$]$_2$ displayed highly promising UV protection and moisture retention properties and, therefore, represents a polyfunctional pigment that is particularly well suited for cosmetic applications.

**Keywords:** titanium oxide; layered titanium phosphate; photocatalysis; optical band gap; sun protective factor; moisture retention

---

## 1. Introduction

Titanium oxide has been widely used as a cosmetic pigment and physical UV-filter for more than 40 years [1]. However, some titanium oxide samples, as well as related materials, exhibit photocatalytic activity that leads to reactive oxygen species (ROS) formation [2–5]. ROS can damage the organic components of cosmetic products and the skin sebum, both of which contribute to irritating inflammatory reactions on the skin surface [6,7]. Therefore, current efforts toward developing new pigments also involve monitoring and suppressing their potential photocatalytic activity.

There are several approaches to suppressing the photocatalytic reactions on the surface of titania, including using amorphous titanium oxide, which has less photocatalytic activity than the crystalline phases [8]. Additionally, photocatalytic activity may be suppressed by doping the titanium oxide with metal ions [9–11] or modifying its surface composition [12–14]. Matsukura and Onoda showed that phosphate modification leads to photocatalytic activity suppression in TiO$_2$-containing pigments [12]. In addition, Onoda et al. [13,14] found that amorphous titanium phosphate possesses less photocatalytic activity than titanium oxide and helps to retain the skin moisture, however, they did not determine the sun-protective factor (SPF) of the samples and thus could not evaluate titanium phosphates as UV-filters.

In the present study, we prepared two nanocrystalline modifications of TiO$_2$; anatase and brookite and crystalline titanium phosphate, by hydrothermal treatment of titanium lactate complex in the

presence of phosphate ions. Moisture retention ability, SPF, and photocatalytic activity of the pure anatase, brookite, and titanium phosphate were quantified and compared with commercially available $TiO_2$ pigments.

## 2. Materials and Methods

The titanium lactate complex was prepared by adding of 17 mL of titanium butoxide (97%, Sigma Aldrich, St. Louis, MO, USA) to a 14 mL solution of lactic acid (85%, Sigma Aldrich) in water. The mixture was heated to 80 °C for about 4 h until the precipitate dissolved and the excess of butanol was evaporated. The final titanium lactate complex solution was diluted to 100 mL. For the hydrothermal synthesis, 4.86 g of urea (95.5% Sigma Aldrich) and 10 mL of the titanium lactate complex solution were combined in deionized water; the total volume of the solution was 24 mL. The final concentration of Ti in the resulting clear solution was 0.0021 M. Ti:P molar ratios ranging from 1:0 to 1:4 were generated by adding different amounts of $NH_4H_2PO_4$ (99%, Fluka, Bucharest, Romania) to the Ti-containing solution. Then the solution was transferred into a Teflon-lined, stainless steel autoclave (total volume 60 mL, 40% loading) and hydrothermally treated at 180 °C for 48 h. The reaction products were washed four times with water and once with ethanol after sequential centrifugations at 7000 rpm and then dried in air at room temperature overnight. The samples were denoted as TiP*n*, where *n* reflects the P:Ti molar ratio.

X-ray diffraction (XRD) patterns were collected using the D/MAX 2500 diffractometer (Rigaku, Tokyo, Japan) with θ–2θ Bragg-Brentano reflection geometry and a scintillation counter. All measurements were performed with CuKα radiation generated by a rotating Cu anode (50 kV, 250 mA). XRD patterns were obtained in the 5–70° 2θ range with a 0.02° step/second. The phase composition of the samples was identified using the ICDD database and the le Bail method was used to fit XRD in the JANA2006 software (Version 25/10/2015, Institute of Physics, Praha, Czech Republic) [15].

The samples structures were analyzed on a NVision 40 (Zeiss, Oberkochen, Germany) high-resolution scanning electron microscope (SEM) at 3 kV equipped with Oxford Instruments X-MAX ($80 mm^2$) detector. EDX analysis was carried out at 20 kV, spectra were acquired from more than 5 points, the data were compared with spectra from area (Table S1 in Supplementary Materials). Transmission electron microscopy images were collected using Leo912 AB Omega (Zeiss, Oberkochen, Germany) TEM.

Raman spectra were acquired using an inVia Raman spectrometer (Renishaw, Wotton-under-Edge, UK) coupled to Leica's DMLM optical microscope through a 50× objective. Measurements were performed at room temperature in the Raman shift frequency range of 100–1460 $cm^{-1}$ using a 50 mW 514 nm argon laser. Before taking the measurement, the Raman spectrometer was calibrated against the $F_{1g}$ line of Si at 520.2 $cm^{-1}$ as a reference. The brookite and anatase bands were assigned according to the literature [16–18].

UV-Vis optical absorption spectra were recorded on a Lambda 950 spectrometer (Perkin Elmer, Waltham, MA, USA) in the 200–1000 nm range using the diffuse reflectance mode. The Kubelka-Munk function for each sample was calculated from reflectance [19]. Calculation of the band gap was described in the literature [20,21].

To measure SPF, 0.1 g of sample was added to 0.9 g of a solution of 90 wt. % water, 9.9 wt. % glycerol (Sigma-Aldrich, 99.5%), and 0.1 wt. % sodium dodecyl sulfate (99%, Sigma-Aldrich). After mixing in an agate mortar 0.1 mL of this suspension was carefully deposited onto the surface of the Vitro-Skin substrate (IMS Inc., Portland, OR, USA). The spectra of the samples prepared were measured on a Lambda 950 spectrometer in the 290–400 nm range, with a 150 mm integrating sphere, and SPF was calculated according to the ISO 24443 standard. The Suntest CPS+ (ATLAS MTS, Mount Prospect, IL, USA) chamber was used to illuminate the samples with UV light.

The procedure used here to assess moisture retention is described in [13]. Briefly, 0.1 g of each sample was mixed with 0.03 g of water and the weight loss was measured under 60% humidity at 20 °C every 20 min during 140 min.

For photocatalytic measurements, an original commercially available quartz reactor of the AceGlass Inc was used. The scheme of the setup is shown in Supplementary Materials Figure S1A. The photocatalytic activity of the samples was measured through the discoloration of $4.4 \times 10^{-5}$ M methyl orange (Sigma-Aldrich, 85%) aqueous solution due to the first stage of the methyl orange degradation results in destruction of a diazo groups inducing a light absorption. During the experiment the reaction mixture was irradiated with UV illumination of the high-pressure Hg bulb (5.5 W). Illumination spectrum of the bulb is presented on the Figure S1B.

Continuous sampling was provided using a peristaltic pump. A flow of the reaction mixture passed through a U-shaped cell, where absorption spectra were collected using an HRX-2000 xenon lamp and a QE65000 spectrophotometer (Ocean Optics, Largo, FL, USA) every 5 s.

The photocatalytic activity was calculated as a first order constant of the reaction of discoloration. In order to make a comparison with other experimental data, the photocatalytic activity of Evonic P25 was also determined. The measurement procedure and calculation are described in detail in our previous work [8].

## 3. Results and Discussion

To study the formation of titanium phosphates under hydrothermal conditions, we varied the concentration of $NH_4H_2PO_4$ in the reaction mixture. In all cases, white powders were obtained. XRD patterns are presented in Figure 1. According to powder XRD, the variation in $NH_4H_2PO_4$ concentration leads to a change in the phase composition of the samples (Figure 1). Without addition of $NH_4H_2PO_4$ a pure brookite phase formed. This result agrees with Kandiel et al. who obtained this phase under similar conditions [17]. For molar ratio Ti:$NH_4H_2PO_4$ = 1:0.5 in the reaction mixture, the anatase phase formed. For higher $NH_4H_2PO_4$ concentrations, both $(NH_4)TiOPO_4$ and $Ti_2O_2H(PO_4)[(NH_4)_2PO_4]_2$ were present in the system. The TiP2 and TiP4 samples contained a single phase $Ti_2O_2H(PO_4)[(NH_4)_2PO_4]_2$, whereas TiP0.75, TiP1, and TiP1.5 contained an $(NH_4)TiOPO_4$ impurity (Figure 1, Table 1). The relative intensity of the $(NH_4)TiOPO_4$ phase reflections decreases along the TiP0.75-TiP1-TiP1.5 series, which reflects an increase in the fraction of $Ti_2O_2H(PO_4)[(NH_4)_2PO_4]_2$ in the reaction products. In previous studies, the $Ti_2O_2H(PO_4)[(NH_4)_2PO_4]_2$ phase also formed in the presence of excess phosphate and ammonium ions [22,23]. Khainakova et al. prepared this phase using $TiCl_4$ as the titanium-containing precursor, $H_3PO_4$ as the P source, and urea as the ammonium source [22]. In [23], $Ti_2O_2H(PO_4)[(NH_4)_2PO_4]_2$ was formed under hydrothermal conditions from amorphous titanium oxide in the presence of $(NH_4)_2HPO_4$. Our results indicate that titanium lactate can also be used as a precursor for $Ti_2O_2H(PO_4)[(NH_4)_2PO_4]_2$ and the synthetic procedure suggested in the present study is very flexible and enables the generation of three pure phases, anatase, brookite, and $Ti_2O_2H(PO_4)[(NH_4)_2PO_4]_2$, by adjusting the phosphate concentration in the correct ratio.

The diffraction pattern of TiP4 was indexed and unit cell parameters were calculated. The unit cell parameters ($a$ = 6.3946(2) Å, $b$ = 10.1160(5) Å, and $c$ = 10.8661(1) Å, space group Pmna) obtained for the phosphate $Ti_2O_2H(PO_4)[(NH_4)_2PO_4]_2$ are in good agreement with the literature [24].

The TiP4 sample, which contained pure $Ti_2O_2H(PO_4)[(NH_4)_2PO_4]_2$, was also characterized by Raman spectroscopy (Figure 2). According to [24], the phase consists of layers which are stacked in the direction of the *b*-axis. The layers consist of $TiO_6$ octahedra and $PO_4$ tetrahedra. There are two different phosphate groups: Orthophosphate and dihydrogen phosphate, in which the protons are substituted by $NH_4^+$ ions. The bands in the Raman spectrum of $Ti_2O_2H(PO_4)[(NH_4)_2PO_4]_2$ were not previously assigned and we used data from [25] for assignment of the peaks in the spectrum of $\gamma$-Ti[$PO_4$][$O_2P(OH)_2$]·$H_2O$ because this compound also consists of layers which are built from $TiO_6$ octahedra and $PO_4$ tetrahedra [25]. The two bands at 178 cm$^{-1}$ and 418 cm$^{-1}$ and shoulder at 234 cm$^{-1}$ can be ascribed to the asymmetric stretching vibrations of the Ti–O groups. The band at 418 cm$^{-1}$ can be attributed to the symmetric stretching vibration of the Ti–O bond. The shoulder at 366 cm$^{-1}$ and the group of bands at 587, 612, and 681 cm$^{-1}$ correspond to the deformation vibrations of the phosphate group. The stretching vibrations of the phosphate group correspond to the bands at 936,

1003, and 1102 cm$^{-1}$, respectively. Finally, the two bands of the ammonium group are also present in the spectrum (1411 and 1675 cm$^{-1}$) [26].

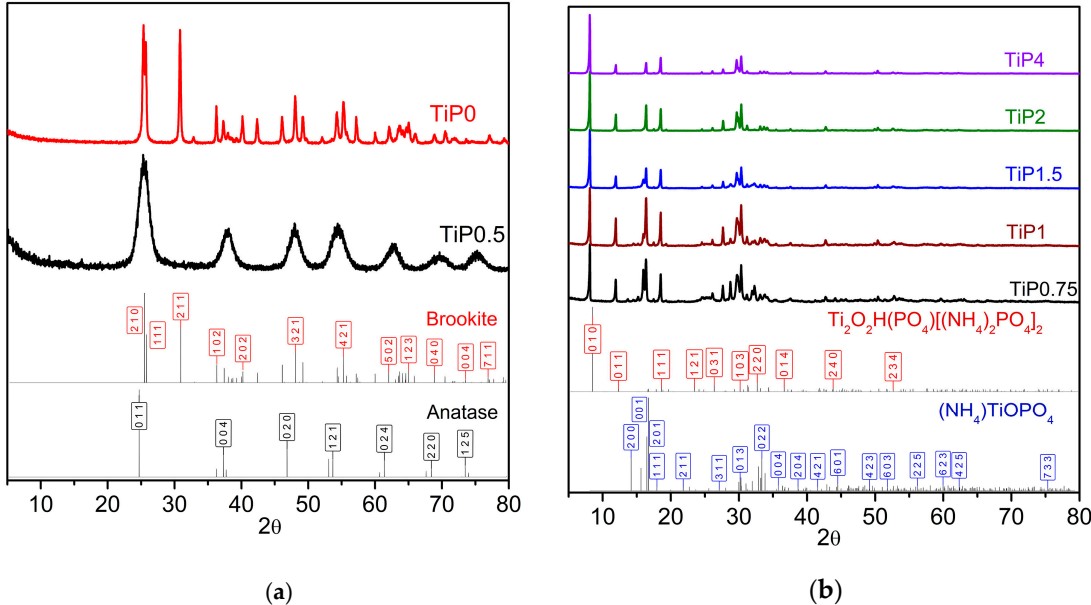

(**a**)　　　　　　　　　　　　(**b**)

**Figure 1.** Powder X-ray diffraction patterns for the samples prepared by hydrothermal treatment of titanium lactate at 180 °C in the presence of urea with different concentrations of phosphate ions; (**a**) XRD patterns of TiP0 and TiP0.5 contain only brookite and anatase respectively, (**b**) XRD patterns of TiP0.75-TiP4 contain mixture of Ti$_2$O$_2$H(PO$_4$)[(NH$_4$)$_2$PO$_4$]$_2$ and (NH$_4$)TiOPO$_4$.

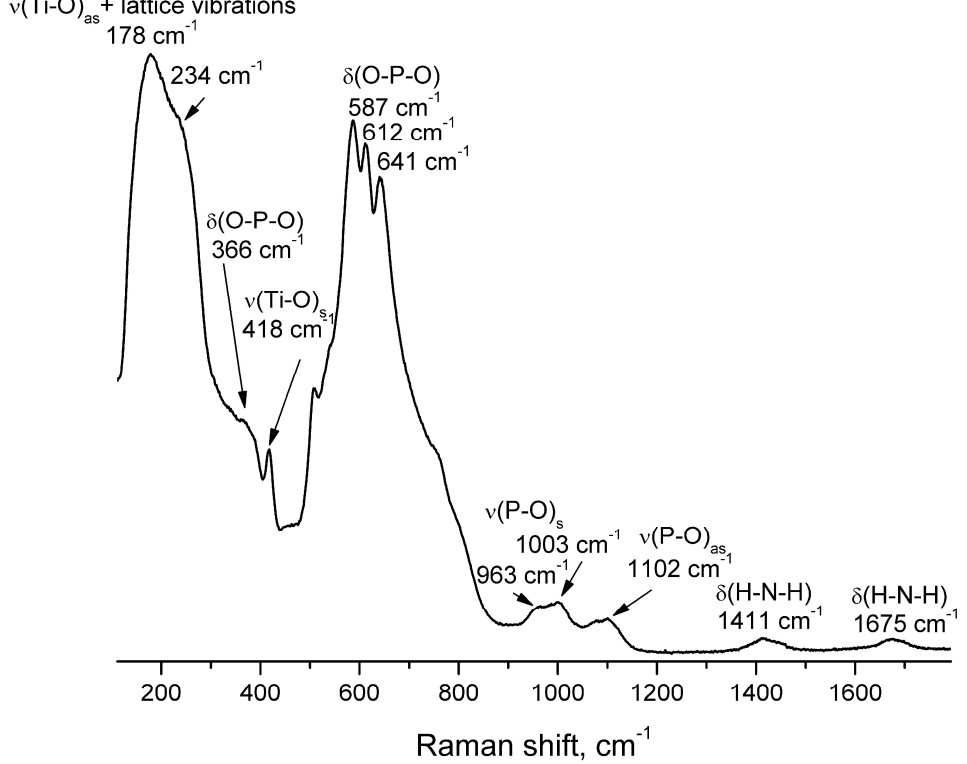

**Figure 2.** Raman spectrum of Ti$_2$O$_2$H(PO$_4$)[(NH$_4$)$_2$PO$_4$]$_2$ prepared by hydrothermally treating titanium lactate at 180 °C in the presence of urea and phosphate ions with a Ti:P ratio of 1:4 (sample TiP4). Bands were assigned according to [25,26].

**Table 1.** The phase compositions and Ti:P molar ratios in the samples prepared by hydrothermally treating the titanium lactate complex at 180 °C in the presence of urea with different concentrations of phosphate ions.

| Sample | Phase Composition | Ti:P Molar Ratio in the Reaction Mixture | Ti:P Molar Ratio in the Sample (EDX) |
|---|---|---|---|
| TiP0 | $TiO_2$ (brookite) | without phosphate | 1:0 |
| TiP0.5 | $TiO_2$ (anatase) | 1:0.5 | 1:0.05 |
| TiP0.75 | $Ti_2O_2H(PO_4)[(NH_4)_2PO_4]_2$ + $(NH_4)TiOPO_4$ | 1:0.75 | 1:1.2 |
| TiP1 | $Ti_2O_2H(PO_4)[(NH_4)_2PO_4]_2$ + $(NH_4)TiOPO_4$ | 1:1 | 1:1.3 |
| TiP1.5 | $Ti_2O_2H(PO_4)[(NH_4)_2PO_4]_2$ + $(NH_4)TiOPO_4$ | 1:1.5 | 1:1.4 |
| TiP2 | $Ti_2O_2H(PO_4)[(NH_4)_2PO_4]_2$ | 1:2 | 1:1.5 |
| TiP4 | $Ti_2O_2H(PO_4)[(NH_4)_2PO_4]_2$ | 1:4 | 1:1.5 |

To get more information about the influence of the increase of Ti:P ratio on the structure and composition the samples were examined using SEM, TEM, and EDX. The results of the EDX analysis are presented in Table 1 and Table S1. In TiP0 no phosphate was determined, and in TiP0.5 phosphate concentration is much lower than in the reaction mixture. For the TiP0.5 sample, this is likely because the phosphate ions only adsorb on the surface of the $TiO_2$ particles and this affects the crystallization process enough to yield an anatase phase, which is in contrast to the TiP0 sample. In the TiP0.75, TiP1, and TiP1.5 samples, the Ti:P ratio is less than 1.5, due to the presence of a second phase, in which the Ti:P ratio is lower than Ti:P = 1:1.5. An increase in the Ti:P ratios determined for the TiP0.75-TiP1-TiP1.5 sample series indicates an increase in the fraction of $Ti_2O_2H(PO_4)[(NH_4)_2PO_4]_2$ in the composition of the samples. From the XRD patterns, the intensity of the reflections of $(NH_4)TiOPO_4$ decrease with the increase in Ti:P ratio. Thus, the EDX data fit nicely with the XRD data. For the TiP2 and TiP4 samples, the Ti:P ratio is 1.5. Therefore, even in presence of excess phosphate ions in the reaction mixture the $Ti_2O_2H(PO_4)[(NH_4)_2PO_4]_2$ phase with a Ti:P ratio of 1.5 is formed.

The morphology of the samples was examined by SEM and TEM. When no phosphate was added to the reaction mixture, rod-like brookite particles formed (Figure 3a,b). This particle shape was observed in [17] for brookite under similar synthetic conditions. TiP0.5 consisted of small aggregated particles around 10 nm in size (Figure 3c,d). Thus, the presence of phosphate ions on the reaction mixture prevented the formation of the brookite phase.

Plate-like particles formed in the samples with Ti:P ratios higher than 0.75 (Figures S2–S6). For all phosphate concentrations, the thickness of the plates was from 0.2 to 0.5 μm and the lateral size was about 3.6–9.3 μm (Table S2). The thickness and lateral sizes of the particles tend to decrease with the increase of phosphate concentration in the reaction mixture. The plate-like morphology observed in $Ti_2O_2H(PO_4)[(NH_4)_2PO_4]_2$ phase was also described in [22]. With increasing concentrations of phosphate ions and decreasing of $(NH_4)TiOPO_4$ content the surfaces seem to become smoother. The samples TiP4 and TiP2, which contain only $Ti_2O_2H(PO_4)[(NH_4)_2PO_4]_2$ phase consist of smooth plate-like particles (Figure 4a,b, Figures S2 and S3), so this phase has a smooth plate-like morphology. In the samples TiP1 and TiP0.75 the plate-like particles are covered by small particles (Figure 4c,d, Figures S5 and S6). According XRD analysis these samples contain $(NH_4)TiOPO_4$ impurity, so $(NH_4)TiOPO_4$ phase consists of small particles which cover $Ti_2O_2H(PO_4)[(NH_4)_2PO_4]_2$ plates. TiP1.5 sample also contains $(NH_4)TiOPO_4$ according XRD, but small particles of this phase are not clearly seen in SEM images (Figure S4), probably, because of low content of this phase in the sample.

Optical properties of the pure phases were studied by diffuse reflectance UV-Vis spectroscopy (Figure 5). To determine the optical band gap, the Kubelka-Munk function $F$ was calculated as follows [19]:

$$F = \frac{(1-R)^2}{2R} \tag{1}$$

where R is reflectance. By plotting $(F \cdot h\nu)^{1/2}$ for indirect allowed transitions vs. h$\nu$ (where h$\nu$ is photon energy), the optical band gaps of the semiconductor samples were obtained. The optical band gap for $Ti_2O_2H(PO_4)[(NH_4)_2PO_4]_2$ was found to be 3.37 eV. It is not clear if $Ti_2O_2H(PO_4)[(NH_4)_2PO_4]_2$ is a direct or indirect semiconductor. The graph was plotted in coordinates $(F \cdot h\nu)^2$ vs. h$\nu$ and the energy of direct transition was found to be 3.53 eV (Figure S7). For the pure brookite and commercial rutile samples, the band gap values agree well with those in the literature [27]. For the commercial anatase and TiP0.5 samples, which also includes anatase, the optical band gaps were found to be at 3.23 and 3.09 eV, respectively. In the literature, the optical band gap for anatase varies from 3.09 to 3.2 eV [27–30]. This discrepancy cannot be explained by the quantum size effect because the exciton radius in anatase is very small [28,29]. However, the formation of energy levels due to defects in anatase crystalline structure may shift the absorption edge and could cause the apparent change of the band gap.

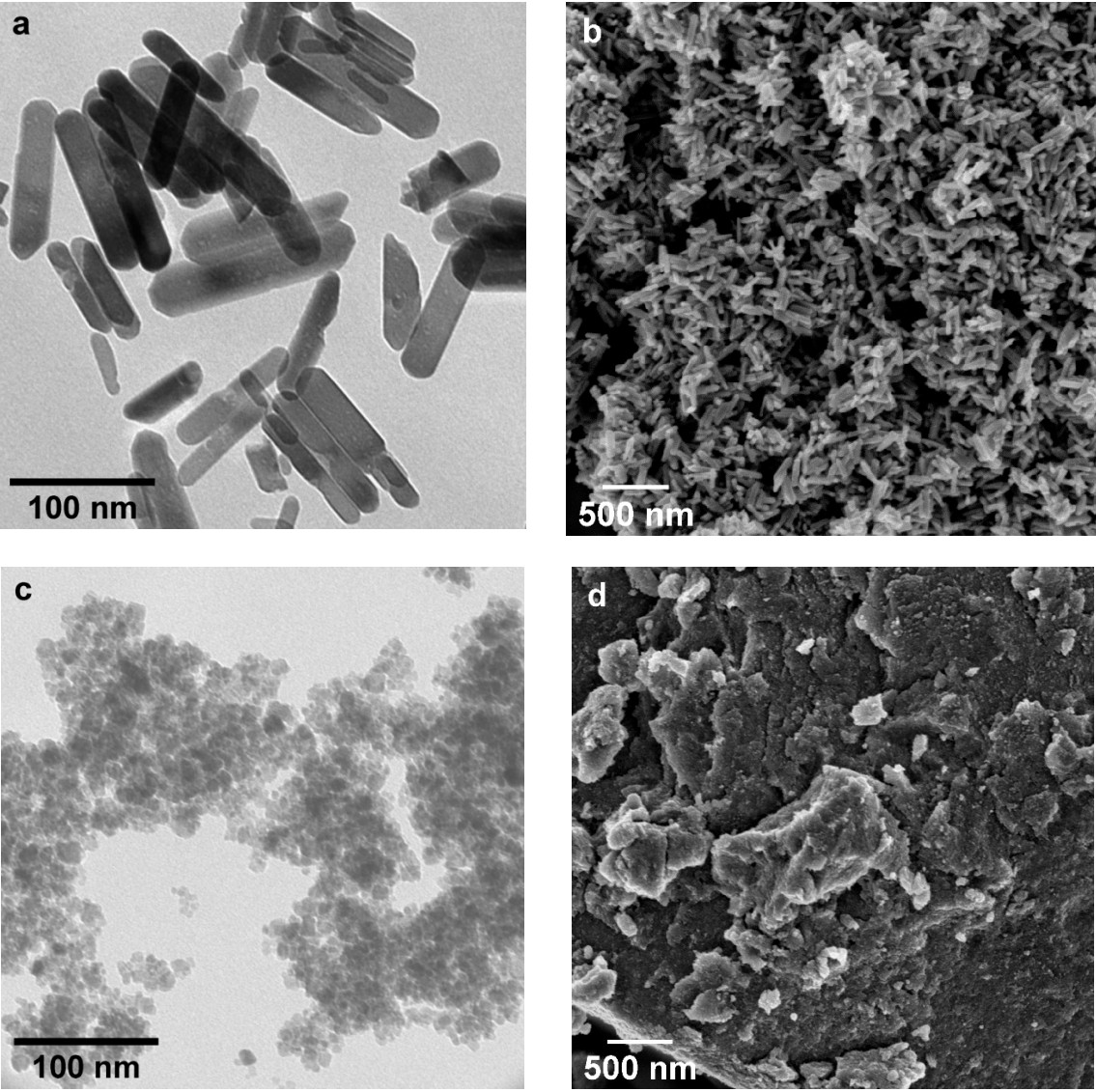

**Figure 3.** The microstructure of the sample, prepared by hydrothermal treatment of titanium lactate complex at 180 °C in the presence of urea without the addition of phosphate (sample TiP0, (**a**) TEM image, (**b**) SEM image); and the microstructure of the sample prepared by hydrothermally treating the titanium lactate complex at 180 °C in the presence of urea with a Ti:P ratio of 1:0.5 in the reaction mixture (TiP0.5, (**c**) TEM image, (**d**) SEM image).

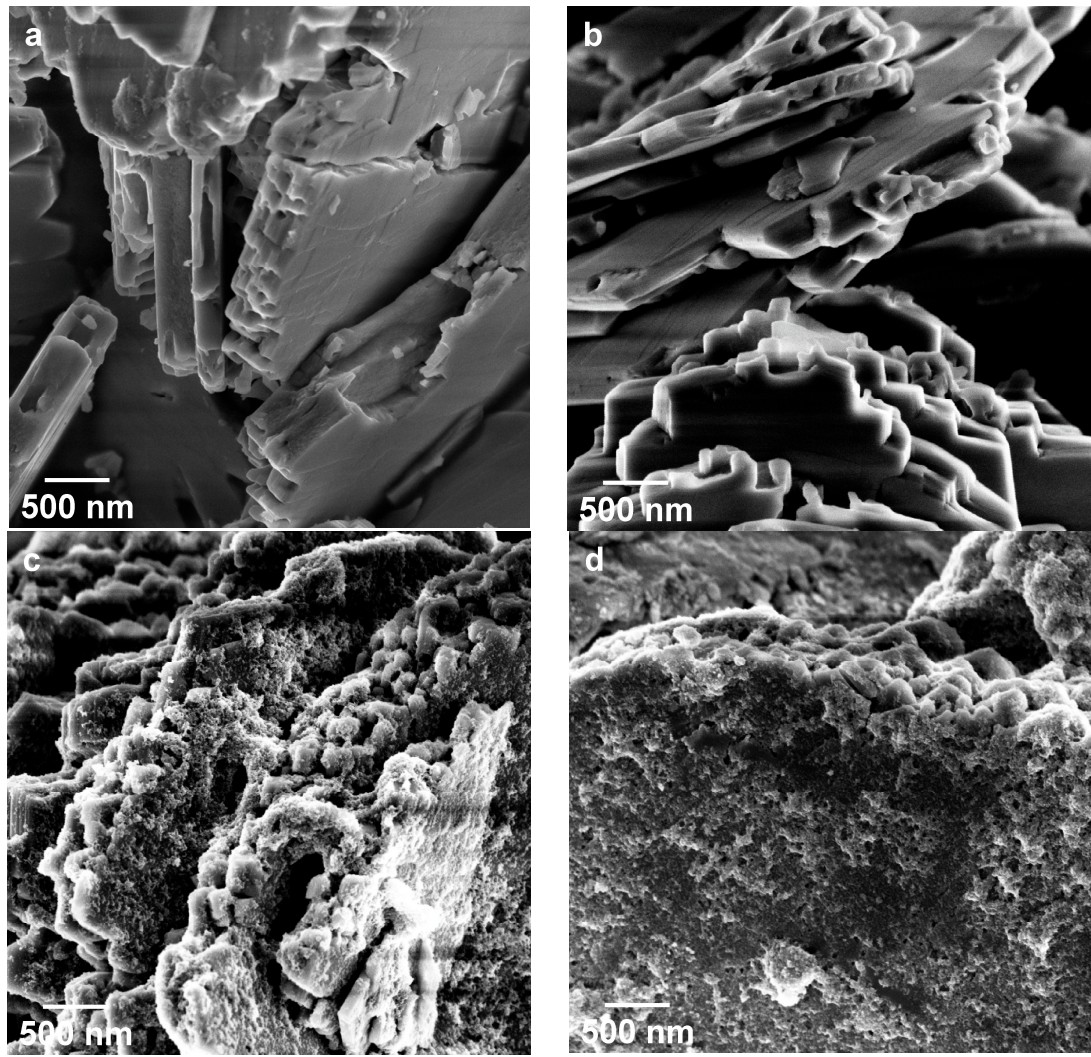

**Figure 4.** The microstructure of the TiP4 (**a**) 4, TiP2 (**b**), TiP1 (**c**), and TiP0.75 (**d**) samples prepared by hydrothermal treatment of titanium lactate complex at 180 °C in the presence of urea and phosphate ions and with Ti:P ratios of 1:4, 1:2, 1:1, and 1:0.75.

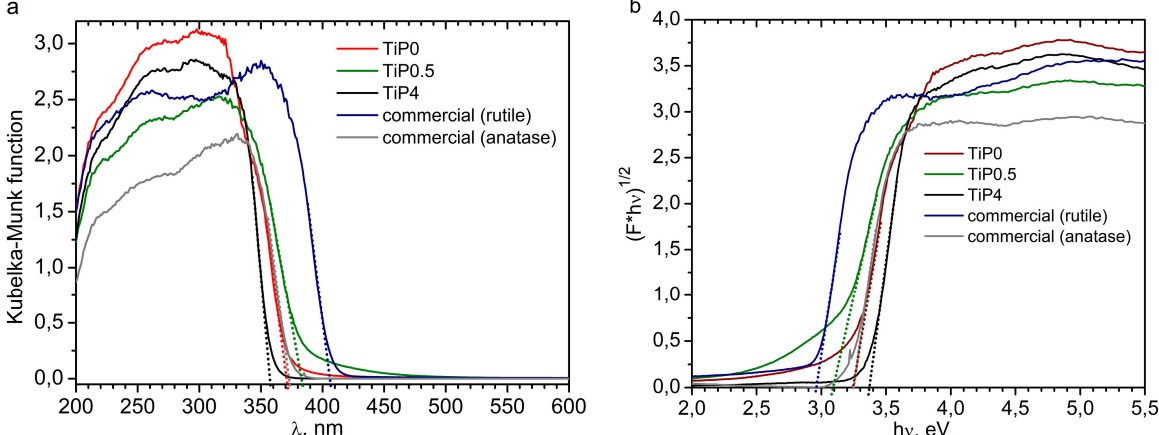

**Figure 5.** Kubelka-Munk function (*F*) vs. wavelength (**a**) and plot of $(F \cdot h\nu)^{1/2}$ vs. $h\nu$ (**b**) for three pure phase samples: TiP0 (brookite), TiP1 (anatase), TiP4 ($Ti_2O_2H(PO_4)[(NH_4)_2PO_4]_2$) and two commercially available samples: anatase and rutile.

The photocatalytic and photoprotective properties of the commercial and prepared within this study samples are presented in Table 2. Commercial cosmetic pigments and nanocrystalline brookite possessed high photocatalytic activity. The TiP0.5 sample was likely less active because it contained amorphous anatase, or its surface may be modified by phosphate groups. According the literature, these factors may decrease photocatalytic activity [8,12]. The $Ti_2O_2H(PO_4)[(NH_4)_2PO_4]_2$ was also photocatalytically active. The photocatalytic activity of crystalline titanium phosphate phases has been previously described ($Ti_2O(PO_4)_2(H_2O)_2$ in [31] and Bi-containing composite with $Ti(HPO_4)_2(H_2O)$ in [32]), however, this is the first report to determine the photocatalytic activity of $Ti_2O_2H(PO_4)[(NH_4)_2PO_4]_2$.

The nanocrystalline brookite (TiP0) possessed the highest SPF of the samples included in this study. However, its photocatalytic activity is also high, so for potential application the suppression of photocatalytic activity is necessary. In contrast, the SPF exhibited by titanium phosphate (TiP4) was the same as the commercial samples and its photocatalytic activity was lower than the commercial samples. Anatase (TiP0.5) displayed the lowest photocatalytic activity, however, its SPF was very low, thus negating its benefit in sun-protective cosmetics. UVAPF (UVA protection factor) values of the samples show the level of protection against UVA part of the spectrum (320–400 nm). The highest value corresponds to brookite sample (TiP0), whereas $Ti_2O_2H(PO_4)[(NH_4)_2PO_4]_2$ does not demonstrate high value. It is clearly seen in the absorption spectra of the samples (Figure S8), that brookite and $Ti_2O_2H(PO_4)[(NH_4)_2PO_4]_2$ phases possess the highest values of protection from UVB rays (290–320 nm), whereas commercial anatase and rutile actively protect from UVA rays. The combination of different phases in cosmetics may provide broad spectrum protection from both UVA and UVB types of solar radiation.

Another advantage of titanium phosphate is its ability to retain moisture. Figure 6 depicts the rate of moisture loss specific to each sample over time. Among all of the samples investigated, titanium phosphate tended to retain the most moisture for the longest period of time. Therefore, it may be superior to traditional titanium oxide. Furthermore, due to its sufficiently high SPF and lower level of photocatalytic activity than brookite and commercial samples, titanium phosphate may be used as multifunctional cosmetic pigment that confers both moisture retention and sun protection.

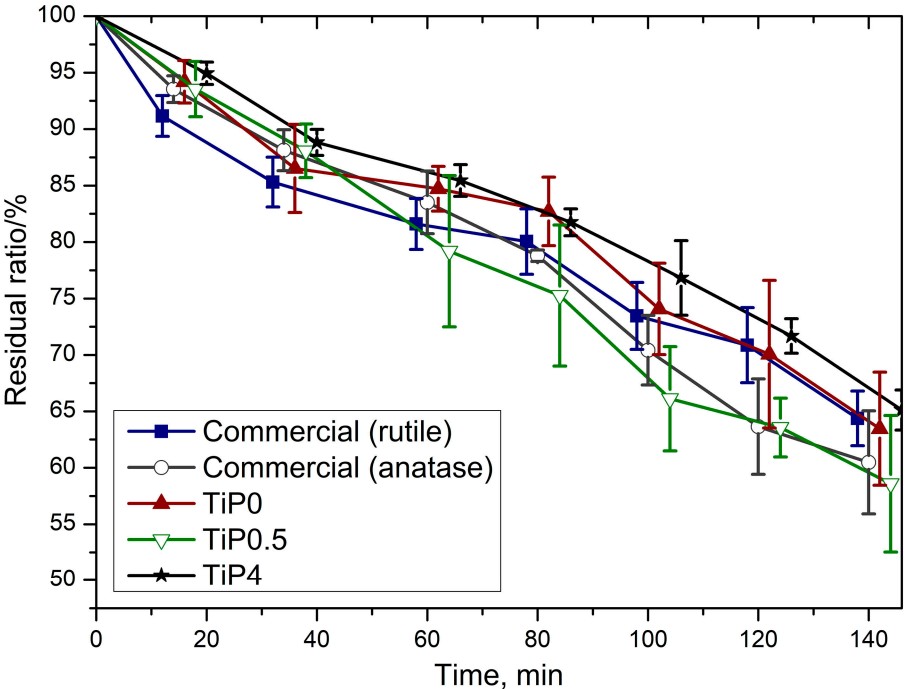

**Figure 6.** Moisture retention in the two commercial samples of $TiO_2$ and the experimental samples prepared in this study: TiP0 (brookite), TiP0.5 (anatase), and TiP4 ($Ti_2O_2H(PO_4)[(NH_4)_2PO_4]_2$).

**Table 2.** Optical band gap, photocatalytic activity, SPF and UVAPF for brookite, anatase, and Table 2. phases (TiP0, TiP0.5, and TiP4 samples, respectively) and for commercially available anatase and rutile samples.

| Sample | Optical Band Gap (eV) | Photocatalytic Activity (% from P25) | SPF | UVAPF |
|---|---|---|---|---|
| TiP0 | 3.25 | 60 | 11 | 5 |
| TiP0.5 | 3.09 | 6 | 1 | 1 |
| TiP4 | 3.37/3.53 [a] | 40 | 3 | 2 |
| Commercial (rutile) | 2.97 | 68 | 3 | 3 |
| Commercial (anatase) | 3.23 | 121 | 3 | 4 |

[a] optical band gap calculated under the assumption that $Ti_2O_2H(PO_4)[(NH_4)_2PO_4]_2$ has an indirect or direct transition, respectively.

## 4. Conclusions

Two crystalline modifications of $TiO_2$, brookite and anatase, as well as layered titanium phosphate $Ti_2O_2H(PO_4)[(NH_4)_2PO_4]_2$ were prepared under hydrothermal conditions. The Ti:P ratio in the reaction mixture determines the phase composition of the resulting complex. In the absence of phosphate ions, a brookite phase was formed. If the Ti:P ratio is 1:0.5, anatase was formed. At higher Ti:P ratios, a mixture between $Ti_2O_2H(PO_4)[(NH_4)_2PO_4]_2$ and $(NH_4)TiOPO_4$ was formed, and at Ti:P ratios higher than 1:2, pure $Ti_2O_2H(PO_4)[(NH_4)_2PO_4]_2$ was obtained. Anatase, brookite, and layered titanium phosphate possessed less photocatalytic activity compared to the commercially available cosmetic pigments. Brookite demonstrated the highest SPF of the studied samples and anatase the lowest, rendering it unsuitable for cosmetic use. $Ti_2O_2H(PO_4)[(NH_4)_2PO_4]_2$ was found to be both UV-protective, at levels comparable with the commercial samples, and moisture-retentive and is therefore an attractive multifunctional pigment for use in the cosmetic industry.

**Supplementary Materials:** The following are available online at http://www.mdpi.com/2073-4352/9/7/332/s1, Table S1: Data of the energy dispersive X-ray analysis; Table S2: particle size parameters for the samples, which contain $Ti_2O_2H(PO_4)[(NH_4)_2PO_4]_2$; Figure S1: (a) Scheme of the photocatalytic measurements setup, (b) Spectrum of the Hg bulb; Figure S2: The microstructure of the TiP4, sample prepared by hydrothermal treatment of titanium lactate complex at 180 °C in the presence of urea and phosphate ions and with Ti:P ratio of 1:4; Figure S3: The microstructure of the TiP2, sample prepared by hydrothermal treatment of titanium lactate complex at 180 °C in the presence of urea and phosphate ions and with Ti:P ratio of 1:2; Figure S4: The microstructure of the TiP1.5, sample prepared by hydrothermal treatment of titanium lactate complex at 180 °C in the presence of urea and phosphate ions and with Ti:P ratio of 1:1.5 (a,b) magnification 50,000 and 5000 respectively; Figure S5: The microstructure of the TiP1, sample prepared by hydrothermal treatment of titanium lactate complex at 180 °C in the presence of urea and phosphate ions and with Ti:P ratio of 1:1; Figure S6: The microstructure of the TiP0.75, sample prepared by hydrothermal treatment of titanium lactate complex at 180 °C in the presence of urea and phosphate ions and with Ti:P ratio of 1:0.75; Figure S7: Plot of $(F \cdot h\nu)^2$ vs. $h\nu$ TiP4 sample ($Ti_2O_2H(PO_4)[(NH_4)_2PO_4]_2$): Eg = 3.53 eV; Figure S8: absorption spectra in UV regions of 10% suspensions of the two commercial samples of $TiO_2$ and the experimental samples prepared in this study: TiP0 (brookite), TiP0.5 (anatase), and TiP4 ($Ti_2O_2H(PO_4)[(NH_4)_2PO_4]_2$).

**Author Contributions:** Conceptualization, I.V.K.; methodology, I.V.K.; preparation of the samples A.N.A., A.S.A., characterization of the samples A.N.A., A.S.A., D.A.K., measurement of properties I.V.K., A.N.A., A.S.A., D.A.K.; writing—original draft preparation, I.V.K., A.N.A., A.S.A.; writing—review and editing, I.V.K., D.A.K.; supervision, I.V.K.

**Funding:** This research was funded by the Russian Science Foundation (grant number 17-73-10493).

**Acknowledgments:** This work was supported by Lomonosov Moscow State University Program of Development. The authors would like to thank S.S. Abramchuk (Faculty of Chemistry, Lomonosov Moscow State University) for the TEM experiments.

**Conflicts of Interest:** The authors declare no conflict of interest.

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
