# Peer review of "Hydrothermal Synthesis of Layered Titanium Phosphate Ti2O2H(PO4)[(NH4)2PO4]2 and Its Potential Application in Cosmetics"

_crystals, doi:10.3390/cryst9070332_

Round 1

Reviewer 1 Report

Minor corrections/changes as annotated in the pdf-file required.

Author Response

Reviewer 1 Line 63 Le Bail Fit?

Answer: yes.

Reviewer 1: Line 68 Please state the accelaration voltages for SEM and for EDX

Answer: Accerelation voltages added.

Reviewer 1: Line 116 The image (Figure 1) is quite poor, both in graphical quality/resolution and in information content. Enlarge the image, indicate the positions of reflection of  brookite and anatase as references for TiP0 and TIP0.5

Answer: Fig. 1 was changed, all required information was added.

Reviewer 1: Line 126 What does that mean: displaced by NH4+ groups?

Answer: The structure of Ti2O2H(PO4)[(NH4)2PO4]2was reported previously [Ref. 24 in manuscript]. More descriptive formula is (NH4)2Ti2O2(PO4)(HPO4)[(NH4)2PO4]. In this layered phosphate the framework possesses 5 negative charges per unit formula. These negative charges can be neutralized by protons or other cations. In this case negative charge is neutralized by 4 ammonium ions and one proton. But ammonium ions can be exchanged to protons like in zeolites or clays. “Displaced” was changed to “substituted”.

Reviewer 1: Line 147 Are different particle shapes/morphologies visible in SEM images? If yes, separate determinations of Ti/ and P contents would be good.

Answer: (NH4)TiOPO4 and Ti2O2H(PO4)[(NH4)2PO4]2 have different morphologies, Ti2O2H(PO4)[(NH4)2PO4]2 has plate-like morphology and (NH4)TiOPO4 may be like smaller particles which cover Ti2O2H(PO4)[(NH4)2PO4]2 plates (see SEM images in Fig. 4 c and d for the samples TiP0.75 and TiP1, in which (NH4)TiOPO4 shows clearly visible reflexes in XRD patterns). In this case it is not possible to do EDX mapping for these phases separately. No other morphologies have been observed.

Reviewer 1: Line 151 I don't understand this statement: How can the product have a Ti:P ratio of 1:1.5 if the starting amounts are 1:1.4? Where is the missing Ti? What does that mean: exceess phosphae in the reaction mixture leads to the formation of...?

Answer: In samples TiP4, TiP2, TiP1.5 the real Ti:P ratio was smaller than Ti:P ratio in the reaction mixture. Excess phosphate stayed in the liquid phase and was removed during washing. In samples TiP1 and TiP0.75 the Ti:P ratio is higher than in the reaction mixture, so titanium could stay in the liquid phase, maybe, due to complex formation. We did not measure Ti and P concentrations in liquid phases after hydrothermal treatment and did not check if titanium complex could form under these conditions.

Reviewer 1: Line 169 I don't see plates in Fig. 4 a.

Answer: plate-like morphology is seen in Fig. S1-S5 in SI.

Reviewer 1: Line 172 What is the reason? (With increasing concentrations of phosphate ions, the surfaces seem to become smoother, due to lower concentrations of (NH4)TiOPO4)

Reviewer 2 Report

Comments

The decrease in photocatalytic activity in amorphous Ti phosphate has been reported previously. What was the motivation of authors to prepare crystalline Ti Phosphates? How is it compared to the photocatalytic activity of amorphous Ti phosphate?

A two-step preparation method is used. Can one expect complete conversion of Ti butoxide to lactate in the first step? The possibility of unreacted Ti butoxide which may cause skin irritation cannot be eliminated.

The procedure for photocatalytic activity is not comprehensive enough. The reference [22] does not give the detailed procedure. The authors much check the reference. Which range of UV radiation is used? UVA or UVB?

In XRD pattern the authors should index all peaks which may give valuable information about the material.

Page 3, line 106-In a previous studies …………..must be changed to …..in our previous studies/study

Page 4-Lines 141 and 142- ‘and in TiP0 determined phosphate concentration is much lower than in the reacton mixture’.The sentence does not make any sense. TiP0 doesn’t contain phosphate. Reacton must be changed to reaction.

EDX is a semi-quantitative technique. Authors needs to clarify how the measurements were done. Single point or multipoint averaging or mapping? Page 4, line 144-146-In the TiP0.75, 145 TiP1, and TiP1.5 samples, the Ti:P ratio is less than 1.5, due to the presence of a second phase, in which the Ti: P ratio is lower than Ti:P = 1:1.5. The sentence needs more clarity. How does the second phase affect the EDX measurement?

SEM image shows flake-like morphology of LDHs. The UV blocking properties of particles also depend on the physical morphology. Authors mentioned thickness of the plates was 0.4 μm and the lateral size was about 5 – 10 μm.Have the authors measured the change in thickness of the flakes with change in concentration of phosphate?

Figs. 1,2, 5 and 6-quality is poor. Authors must provide high quality graphs so that readers readers can identify the properties of specific sample easily. On Figure 5,   (a), (b) labels are missing. 5 (b)-Y axis label must be corrected (hn to hν). Page 8, figure must be Figure 6. Correction needed in the text as well.

Anatase is widely reported as highly photocatalytic than Brookite. What could be the reason for the lower photocatalytic activity of anatase in this case? The sentence- The TiP0.5 sample was likely less active because it contained nanocrystalline anatase and defects in the crystalline lattice may trap the charge carriers’ needs more explanation.

Moreover, changes are observed among various samples in their capacity for UVA and UVB absorption. It would be more informative if the discussion part incudes these observations as well.

Authors mentioned that Photocatalytic activity of TiP4 is 40% from P25?  Did authors assume P25 activity as 100%? What were the conditions of UV irradiation? It needs clarification.

What is the refractive index of the material? The particle size seems to be in the micron range. How will this affect aesthetic appearance of cosmetic formulations?

Polyfunctional should be changed to multifunctional

The objective of the study is to reduce the photocatalytic activity of the sunscreen additive. Is 40% photoreactivity of TiP4 compared to P25 good enough to eliminate the harmful effects of reactive oxygen species? Moreover, this sample demonstrated low SPF.

How many repeats were done for the moisture retention measurements? Authors must provide statistical data with error bar. Single point measurement is not reliable and acceptable.

Author Response

Point 1:

The decrease in photocatalytic activity in amorphous Ti phosphate has been reported previously. What was the motivation of authors to prepare crystalline Ti Phosphates? How is it compared to the photocatalytic activity of amorphous Ti phosphate?

Answer 1: It was shown that crystalline titanium phosphates demonstrated low photocatalytic activity (Ti(HPO4)2(H20) in 10.1016/j.jcis.2016.05.021, Ref. 32  and Ti2O(PO4)2(H2O)2 in 10.1016/j.jphotochem.2017.01.016? Ref. 33). Photocatalytic activity of Ti2O2H(PO4)[(NH4)2PO4]2 was not studied before. In the beginning of our study we supposed that its activity will be low. And we have found that it is lower than photocatalytic activity of commercial TiO2 photocatalyst (Degussa P25) and then photocatalytic activity of commercial cosmetic TiO2 pigments. But it is non-zero, so it can produce free radicals under UV irradiation.

Amorphous titanium phosphate has low photocatalytic activity [10.4236/msa.2012.31003, Ref. 13], but it is impossible to compare literature data and the data in this paper in absolute values because the experiments were performed in different conditions.

Point 2:

A two-step preparation method is used. Can one expect complete conversion of Ti butoxide to lactate in the first step? The possibility of unreacted Ti butoxide which may cause skin irritation cannot be eliminated.

Answer 2: At the first stage titanium butoxide was added to water solution of lactic acid and white precipitate was formed. In our opinion this precipitate was hydrated titanium oxide, because titanium butoxide like other alcoxides is hydrolyzed rapidly and irreversibly [10.1016/0079-6786(88)90005-2]: Ti(OC4H9)4 + (2+x) H2O → TiO2*xH2O + 4 HOC4H9.

Ti

CH3–HC–O

O=C–O

O–CH–CH3

O–C=O

O–CH–CH3

O–C=O

The precipitate was stirred together with lactic acid and butanol at 80°C in open beaker and after 4 hours it dissolved due to the reaction with lactic acid and the formation of titanium lactate complex:

TiO2*xH2O + CH3CH(OH)COOH → [                                              ]2– + 2H+ + (2+x)H2O

In fact, the structure of titanium lactate complex in these conditions may be different, for example trimers of tetramers may be formed, and hydroxyl groups may coordinate to titanium atom as well. We did not study exact structure of titanium lactate complex. Anyway, titanium butoxide cannot exist under these conditions.

At the same time while the suspension was heated, butanol was partly removed by evaporation, however we did not check if the traces of butanol were present in the solution before hydrothermal treatment.

At the second stage the solution which contained titanium lactate complex, urea and ammonium dihydrophosphate was hydrothermally treated. Titanium butoxide could not be formed in these conditions because it hydrolyses easily. The precipitates which formed after hydrothermal synthesis were washed several times. So, there cannot be titanium butoxide in final powders.

Point 3:

The procedure for photocatalytic activity is not comprehensive enough. The reference [22] does not give the detailed procedure. The authors much check the reference. Which range of UV radiation is used? UVA or UVB?

Answer 3: Reference was changed, the details added to the manuscript and SI.

Point 4:

In XRD pattern the authors should index all peaks which may give valuable information about the material.

Answer: All peaks of the phases were indexed. XRD patterns were compared with database data. Information added to the manuscript. Only the strongest peaks indexes are shown at the graphs.

Point 5:

Page 3, line 106-In a previous studies …………..must be changed to …..in our previous studies/study

Answer: changed

Point 6:

Page 4-Lines 141 and 142- ‘and in TiP0 determined phosphate concentration is much lower than in the reacton mixture’.The sentence does not make any sense. TiP0 doesn’t contain phosphate. Reacton must be changed to reaction.

Answer: changed

Point 7:

EDX is a semi-quantitative technique. Authors needs to clarify how the measurements were done. Single point or multipoint averaging or mapping? Page 4, line 144-146-In the TiP0.75, 145 TiP1, and TiP1.5 samples, the Ti:P ratio is less than 1.5, due to the presence of a second phase, in which the Ti: P ratio is lower than Ti:P = 1:1.5. The sentence needs more clarity. How does the second phase affect the EDX measurement?

Answer:

EDX analysis was acquired in 5 points and averaged, and from area 100*150 µm. Averaged data were added to SI, Table S1.

Point 8:

SEM image shows flake-like morphology of LDHs. The UV blocking properties of particles also depend on the physical morphology. Authors mentioned thickness of the plates was 0.4 μm and the lateral size was about 5 – 10 μm. Have the authors measured the change in thickness of the flakes with change in concentration of phosphate?

Answer:

We have measured the thickness of the plates with the change in concentration of phosphate ions in the reaction mixture: it tended to decrease with the increase of phosphate concentration. Assuming plates of Ti2O2H(PO4)[(NH4)2PO4]2 as rhombs with diagonals d1 and d2, we also observed that average size of the plates decreased. Table S2 with estimated sizes of Ti2O2H(PO4)[(NH4)2PO4]2 particles was added to SI.

Point 9:

Figs. 1,2, 5 and 6-quality is poor. Authors must provide high quality graphs so that readers readers can identify the properties of specific sample easily. On Figure 5,   (a), (b) labels are missing. 5 (b)-Y axis label must be corrected (hn to hν). Page 8, figure must be Figure 6. Correction needed in the text as well.

Answer: figures were changed.

Point 10:

Anatase is widely reported as highly photocatalytic than Brookite. What could be the reason for the lower photocatalytic activity of anatase in this case? The sentence- ‘The TiP0.5 sample was likely less active because it contained nanocrystalline anatase and defects in the crystalline lattice may trap the charge carriers’ needs more explanation.

Answer: The data on photocatalytic activity of brookite are contradictory, it can be more active or less active then anatase (doi:10.3390/catal7100304, doi:10.3390/catal3010036). Low photocatalytic activity of anatase obtained in our work can be explained by modification of the surface by phosphate (like in Ref. 12) and/or by the presence of XRD-amorphous phase which decreases photocatalytic activity of TiO2 (Ref. 8 in the manuscript).

Point 11:

Moreover, changes are observed among various samples in their capacity for UVA and UVB absorption. It would be more informative if the discussion part incudes these observations as well.

Answer:

The discussion was added.

Point 12:

Authors mentioned that Photocatalytic activity of TiP4 is 40% from P25?  Did authors assume P25 activity as 100%? What were the conditions of UV irradiation? It needs clarification.

Answer:

More details were added to Expreimental and SI about measurements of photocalalytic activity.

Point 13:

What is the refractive index of the material? The particle size seems to be in the micron range. How will this affect aesthetic appearance of cosmetic formulations?

Answer:

The measurement of the refractive index of powders is a complicated task. In this work we did not measure refractive index of Ti2O2H(PO4)[(NH4)2PO4]2 and did not find it in the literature. Boron nitride particles with sheet-like morphology and several microns in diameter are used in mineral cosmetic (powdered eyeshadows and foundations) as fillers and provide beautiful skin appearance due to soft focus effect.

Point 14:

Polyfunctional should be changed to multifunctional

Answer:

Changed.

Point 15:

The objective of the study is to reduce the photocatalytic activity of the sunscreen additive. Is 40% photoreactivity of TiP4 compared to P25 good enough to eliminate the harmful effects of reactive oxygen species? Moreover, this sample demonstrated low SPF.

Answer:

The sample TiP4 has the same SPF as commercial powders. Also, it has lower photocatalytic activity then commercial samples (68% and 121% comparing to P25). This is its benefit in comparison with commercial samples.

Point 16:

How many repeats were done for the moisture retention measurements? Authors must provide statistical data with error bar. Single point measurement is not reliable and acceptable.

Answer:

4 repeats were measured for each point. New graph with errors was added.

Round 2

Reviewer 2 Report

The authors revised the manuscript as per the suggestions. The manuscript may be accepted for publication. However, it needs moderate English language changes and spell check.

Table 2 (Supporting information)- Better to present average values with standard deviation 

Figure 8 (Supporting information)-The colour codes must be presented with labels to identify the samples

Author Response

Point 1:

Table S2 (Supporting information)- Better to present average values with standard deviation 

Answer: Standard deviation were added

Point 2:

Figure S8 (Supporting information)-The colour codes must be presented with labels to identify the samples

Answer: color codes were added

Also English language and spelling was checked.

Crystals EISSN 2073-4352 Published by MDPI AG, Basel, Switzerland RSS E-Mail Table of Contents Alert
Back to Top